# Neglected Tropical Diseases

# Factors associated with health worker adoption of facial and environmental hygiene promotion in the 'SAFE strategy' for trachoma elimination in Western Province, Zambia

**Martha Kasongo** [1]*, **Choolwe Jacobs**[1], **Adam Silumbwe**[2], **Patricia Maritim**[2], **Joseph Mumba Zulu**[2,3], **Hikabasa Halwindi**[4]

**1** Department of Epidemiology and Biostatistics, School of Public Health, University of Zambia, Lusaka, Zambia, **2** Department of Health Policy and Management, School of Public Health, University of Zambia, Lusaka, Zambia, **3** Department of Health Promotion and Education, School of Public Health, University of Zambia, Lusaka, Zambia, **4** Department of Community and Family Medicine, School of Public Health, University of Zambia, Lusaka, Zambia

\* mkasongo10@gmail.com

## Abstract

### Background

Trachoma is responsible for the blindness or visual impairment of about 1.9 million people and causes about 1.4% of all blindness worldwide. In Zambia, trachoma is endemic and Western Province is one of the most affected provinces. The SAFE (surgery, antibiotics, facial cleanliness and environmental improvement) strategy is recommended for elimination of trachoma. In many settings, interventions particularly for facial cleanliness and environmental improvement are sub-optimally adopted due to lack of prioritization and inadequate funding of intervention activities. This study sought to establish the level of, and factors associated with adoption of facial and environmental hygiene promotion in the SAFE strategy among health workers in Western Province, Zambia.

### Methodology/principal findings

This was a cross-sectional study involving 24 health facilities selected from three districts using stratified random sampling. A total of 388 health workers comprising environmental health officers, community health assistants and community health workers were randomly selected. Adoption of facial and environmental hygiene promotion was self-reported, defined as participation in community distribution of information, education and communication (IEC) materials or community demonstrations of correct hand and face washing methods or both, within the past six months. Multiple logistic regression was used to identify the factors associated with adoption using STATA Version 15. The study was conducted in March and April 2023.

**Data availability statement:** All relevant data are within the manuscript and its Supporting Information files.

**Funding:** MK. received support as a recipient of a TDR scholarship under the Postgraduate Training Scheme in Implementation Research at the University of Zambia. We are grateful for the support for the training scheme provided by UNICEF, the UNDP, the World Bank and the WHO Special Program for Research and Training in Tropical Diseases (TDR). The funders had no role in study design, data collection and analysis, decision to publish or preparation of the manuscript.

**Competing interests:** The authors have declared that no competing interests exist.

Adoption of facial and environmental hygiene promotion was low at 47.68%. Having readily available transport (AOR = 3.06. 95% CI = [1.38, 6.80]), perceiving the intervention as relevant for trachoma prevention (AOR = 7.78, 95% CI = [4.38, 13.82]), having been trained in F and E (AOR = 2.17, 95% CI = [1.24, 3.78]) and availability of information, education and communication materials (AOR = 3.04, 95% CI = [1.69, 5.46]) were associated with higher odds of adoption of facial and environmental hygiene promotion among health workers.

### Conclusion/significance

There was low adoption of facial and environmental hygiene promotion among health workers influenced by training, transport availability, IEC material availability and perceived relevance and complexity of the intervention. To increase adoption of facial and environmental hygiene promotion, program implementers must ensure that they consider the identified factors in the planning of the intervention activities.

### Author summary

Trachoma remains a public health problem in many low-income countries including Zambia. WHO recommends the surgery, antibiotics, facial cleanliness and environmental improvements (SAFE strategy) for elimination of trachoma in endemic communities. Studies have shown that facial and environmental hygiene promotion has not been adequately adopted in many places and the associated factors are unclear. We determined the level of, and factors associated with adoption of facial and environmental hygiene promotion among health workers in three districts in Western Province, Zambia. We found that adoption of the intervention was generally low (47.68%) and was highest in Kalabo District compared to the other districts. The most widely adopted core component of the intervention was community demonstrations of correct hand and face washing methods. Participants who had available transport, perceived the intervention as relevant for trachoma prevention, had received training in F and E components of the SAFE strategy and those who had IEC materials were more likely to adopt the intervention. To enhance health worker adoption of facial and environmental hygiene promotion, trachoma elimination program officers and stakeholders should ensure that they consider these factors during the planning and implementation of the SAFE strategy.

### Introduction

Trachoma mostly affects communities with inadequate access to water and sanitation [1]. The disease is responsible for the blindness or visual impairment of about 1.9 million people and causes about 1.4% of all blindness worldwide [1,2]. Approximately, 103 million people in the world live in trachoma endemic areas and are at risk of

trachoma blindness [1]. In Africa, trachoma is endemic. The prevalence of Trachomatous inflammation-follicular (TF), an eye disease caused by repeated infection with *Chlamydia trachomatis*, is greater than 5% in children aged 1–9 years old in 44 countries. Of these, 26 countries are in sub-Saharan Africa, accounting for 85% of the global trachoma burden [3].

The World Health Organization (WHO) recommends the SAFE (surgery, antibiotics, facial cleanliness and environmental improvement) strategy for the elimination of trachoma [4]. In this strategy, surgery is used to stop trichiasis cases from leading to blindness, antibiotics are used to treat active infections and prevent further transmission and facial cleanliness and environmental improvements are used to prevent transmission of the parasite by flies [1,5]. Evidence shows that improvements in facial cleanliness and environmental hygiene have significantly contributed to reduction in the prevalence of trachoma in endemic communities [6].

Despite the importance of facial cleanliness and environmental improvements, there is evidence that demonstrates that these interventions are inadequately implemented in some endemic settings and this could explain the slow progress towards elimination of trachoma in certain endemic countries [7]. For example, a study in Ethiopia found that there was inadequate implementation of facial and environmental hygiene promotion activities as only 56% and 68% of SAFE implementing districts implemented and reported on "F" and "E" components respectively [8]. In addition, the adoption of facial and environmental hygiene promotion is reported to be affected by health workers' attitudes and limited knowledge on the importance of facial cleanliness and environmental hygiene for trachoma elimination [9,10].

In Zambia, trachoma is a major public health problem. According to the Ministry of Health (MoH), the prevalence of trachomatous inflammation-follicular is more than 5% in 1–9 year olds in 16 of the districts and nearly 1.5 million people are at risk of infection [11,12]. Although a recent survey has shown a reduction in the prevalence, trachoma remains a public health problem in Zambia [13].

In 2013, Zambia through the MoH and supporting non-governmental organizations such as Lions Aid Norway started implementing the full SAFE strategy for trachoma elimination in endemic districts. To facilitate the implementation process, the MoH developed the Zambia SAFE Strategy Implementation Framework which provided guidelines on intervention activities. Facial and environmental hygiene promotion was done by health workers in collaboration with community health workers to facilitate community-wide adoption of facial cleanliness and environmental hygiene practices [14]. The core components of facial and environmental hygiene promotion comprised community sensitization using information, education and communication (IEC) material and demonstrations of correct hand and face washing methods [14]. Other activities included stakeholder consultation meetings, promotion of these interventions in schools and engagement of community leaders to champion change in their communities [14].

Like other low-income countries, little is known about the factors associated with adoption of facial and environmental hygiene promotion in Zambia as no study has investigated the F and E components of the SAFE strategy in the country. Similar studies conducted in other countries have assessed implementation of the SAFE strategy interventions, but it is unclear if their findings are generalizable to Zambia [8,15]. Understanding the factors associated with adoption of facial and environmental hygiene promotion can guide trachoma elimination programs in developing strategies to strengthen adoption and implementation of these interventions, thereby accelerating trachoma elimination [16]. This study therefore sought to establish the level of, and factors associated with adoption of facial and environmental hygiene promotion among health workers in Western Province, Zambia.

## Methods

### Ethics statement

This study did not involve child participants. The study participants were health workers, all aged 18 years and older. All participants provided formal written informed consent in their preferred language (Lozi or English). The University of Zambia Biomedical Research Ethics Committee (REF.No. 3469–2022) and the National Health Research Authority approved this study, and permission was obtained from the provincial health office, district health offices as well as health facility

in-charges. Confidentiality and anonymity were maintained by ensuring that the data collected did not contain personal identifiers and the data was stored on password protected gadgets.

## Conceptual or theoretical framework

This study employed the RE-AIM (reach, effectiveness, adoption, implementation and maintenance) framework [17], which conceptualizes the public health impact of an intervention as a function of its implementation outcomes and the context. According to RE-AIM, adoption refers to the initial decision or action to try or employ an innovation or evidence-based practice [17,18]. We focused on adoption as it is one of the understudied implementation outcomes with regards to the 'F' and 'E' in the SAFE Strategy. Although the RE-AIM framework has been used in previous trachoma studies, many have largely concentrated on assessing reach and effectiveness of trachoma interventions [19,20]. No studies have attempted to assess and identify the contextual factors shaping adoption of trachoma control interventions among health providers. A systematic review of interventions used the RE-AIM framework to identify effective and ineffective strategies for addressing barriers to the uptake of eye care services in developing countries [19]. Another study in Niger used the RE-AIM framework to assess reach of azithromycin distribution for child survival by comparing door to door and fixed delivery strategies [20].

## Study design

This was a cross-sectional study. This design was ideal for determining the proportion of health workers who had adopted facial and environmental hygiene promotion while allowing the identification of the factors associated with adoption of the intervention [21].

## Study setting

This study was conducted in Shangombo, Kalabo and Kaoma districts in Western Province of Zambia in the months of March and April 2023. At the time of the study, Western Province had the highest prevalence of trachoma with some districts having as high as 17% prevalence of TF in children aged 1–9 years old [11,12]. Seven districts were implementing the full SAFE strategy in Western Province at the time of the study. Of these, only three districts met the inclusion criteria of having commenced implementation of the SAFE strategy in 2013 and having a trachoma prevalence (TF prevalence in children aged 1–9 years old) greater than 10% as of 2018 [11,14]. At the time of the study, Western Province had approximately 307 health facilities that were offering primary health care services. Of these, approximately 52 were distributed in the three study districts. Data was collected in primary health care facilities including urban health centers, rural health centers and health posts.

## Sampling and participant recruitment

The study population included environmental health technologists, community health assistants and community health workers. In the three districts, the total population was estimated at 564 health workers (512 community health workers and 52 environmental health technologists and community health assistants). The sample size was calculated using Cochran's equation, $n \geq (Z^2 p [1-p])/e^2$ where, n is the minimum estimated sample size, z is the selected critical value of the desired confidence level (1.96 for 95% confidence level); p is the estimated proportion of an attribute that is present in the population (assumed to be 0.50 given the lack of a previous study from which to estimate the population proportion); and e is the desired level of precision settled at 5%. Using the given formula, a sample size of 384 workers was calculated. However, a sample size of 390 was used to allow a 2% non-response rate. Using probabilities proportional to size of the number of health workers in each district, the random sample sizes for each district were calculated as 166, 60 and 164 for Kaoma, Shangombo and Kalabo, respectively.

A total of 24 health facilities were randomly selected from a pool of 52 using probability proportional to size (PPS). Within each district, facilities were stratified by type including urban health centers, rural health centers, and health posts, and listed accordingly. A PPS-based random sample was then drawn from each stratum. Similarly, the number of health workers sampled from each facility was estimated using PPS based on the total number of health workers at each facility. Finally, systematic sampling was used to select participants at the health facility using a sampling frame provided by the facility in-charge. This was done by selecting every $n^{th}$ person, calculated by dividing the total number of health workers listed on the sampling frame by the required sample size at a particular health facility. In a case where a participant did not give consent or was unreachable, the next name on the list was selected. Health workers were eligible to participate in the study if they were involved in community health promotion activities and had worked in their position for not less than one year.

## Data collection

Data was collected using a structured questionnaire using an online tool, Kobo Collect. The questionnaire was prepared in English and verbally translated to the local language, Lozi, during the data collection. The principal investigator collected the data with the help of 3 research assistants who were natives of the three districts included in the study. These research assistants received training on the study and data collection tool a week prior to the commencement of data collection. The questionnaire was piloted in separate health facilities. The data was collected over a period of one month.

## Measurement of variables

As discussed earlier, the dependent variable was adoption of facial and environmental hygiene promotion for trachoma elimination. Adoption was measured on a binary scale. Health workers were considered to have adopted facial and environmental hygiene promotion if they had conducted or taken part in community distribution of IEC materials or community demonstrations of correct hand and face washing methods or both in the past six months. All the independent variables or contextual factors influencing adoption were measured on the nominal scale, except for age which was a continuous variable. The independent variables included socio-demographic factors (age, sex and district), facility type, training, availability of IEC material, availability of transport, perceived intervention complexity, perceived intervention relevance and attitude towards facial and environmental hygiene promotion.

Training was a dichotomous variable coded as yes if a participant had received training on F and E components of the SAFE strategy. IEC material availability was dichotomized, denoting presence or absence of IEC materials containing information on the 'F' and 'E' components of the SAFE strategy. This could be any type of IEC material that an individual health center might have made or obtained from other trachoma stakeholders or through any other means. Transport was classified as a categorical variable, categorized as readily available if it was accessible at least once per week, rarely available if accessible at least once per month but not weekly, and not available if it was either entirely absent or unavailable for a month or more. Perceived complexity of the intervention was measured as a dichotomous variable, based on whether respondents found facial and environmental hygiene promotion activities complicated and difficult to carry out. Similarly, perceived relevance was a dichotomous variable, determined by whether respondents believed that facial and environmental hygiene promotion was important for preventing trachoma transmission.

Lastly, attitude towards facial and environmental hygiene promotion was assessed by asking participants how they felt about the statement: "There is no need for me to conduct facial and environmental hygiene promotion activities in a community where mass drug administration (MDA) has recently been conducted." Response options included: strongly agree, agree, neutral, disagree, and strongly disagree, which were considered as reflecting attitudes ranging from very negative to very positive.

## Data analysis

During data analysis, the distribution of the continuous variable was checked for normality using a histogram with a super imposed normal curve. The continuous variable, age, was not normally distributed and therefore, we reported the median

and inter-quartile range. To test associations between categorical variables, Pearson's chi square test was used upon satisfying the assumptions. Otherwise, Fisher's exact test was used. The non-parametric, Wilcoxon rank sum test was used to check for the association between the outcome variable and the continuous variable, age.

An investigator led backward stepwise multivariable logistic regression was used. Both binary and multivariable logistic regression were performed to identify factors associated with adoption of facial and environmental hygiene promotion. Firstly, binary logistic regression was performed on all independent variables against the outcome variable. In the second step, we run a multivariable logistic regression containing all the variables that were significant at binary logistic regression analysis. After this, the variables with the highest p-values were individually removed from the model, retaining only significant variables that best explain the data (best fit model). However, given the stratified random sampling design used in this study, we incorporated both the strata and cluster variables regardless of whether they were significant or not, into the model, to avoid underestimating the confidence intervals. Selection of the best model was performed using the likelihood ratio test. We reported robust estimates of odds ratios and confidence intervals at both binary and multivariable logistic regression to counter any possible clustering effects. Variables were declared statistically significant at $p < 0.05$. Data was analyzed using STATA software version 15.0 SE (Stata Corporation, College Station, TX, USA).

## Results

### Characteristics of the participants

A total of 388 health workers completed the questionnaire. Just over half were female 50.77% (197/388), with a median age of 40 years (IQR: 34–50). A majority 51.55% (200/388) were from Rural Health Centers. Over half 52.06% (202/388) considered facial and environmental hygiene promotion relevant for trachoma prevention, yet 56.19% (218/388) had not received F and E training. About a third 33.25% (129/388) expressed a very positive attitude toward conducting facial and environmental hygiene promotion activities following recent MDA. Factors such as district, supervisory role, transport availability, perceived complexity and relevance, training, IEC material availability, and attitude were all associated with the adoption of facial and environmental hygiene promotion (Table 1).

### Level of adoption of facial and environmental hygiene promotion

Out of the 388 participants, only 47.68% (185/388) of the respondents had adopted facial and environmental hygiene promotion. Of the three districts, Kalabo District had the highest proportion, 58.90% (96/163) of participants who had adopted facial and environmental hygiene promotion. Shangombo had the least proportion of respondents who adopted the intervention (Fig 1).

The most widely adopted strategy was community leader engagement for promotion of facial cleanliness and environmental improvements, with 65.46% (254/388) of health workers engaging community leaders when conducting intervention activities. On the other hand, the least adopted strategy was community health education using IEC materials with only 24% (89/388) participating in it (Fig 2). Community demonstrations of correct hand and face washing methods were widely adopted in Kaoma, 35.20% (58/165) and Kalabo, 52.10% (85/163).

### Factors associated with adoption of facial and environmental hygiene promotion

Results of univariate analysis showed that 10 variables were significantly associated with adoption, and 8 remained significant in the final multivariable logistic regression model (Table 2). Participants who were from Kaoma (AOR = 0.39, 95% CI = [0.20, 0.76]), and Shangombo (AOR = 0.36, 95% CI = [0.16, 0.81]) were significantly less likely to adopt the intervention compared to those who were from Kalabo. On the other hand, participants who had readily available transport (AOR = 3.06, 95% CI = [1.38, 6.80]), who perceived the intervention as not being complicated (AOR = 2.54, 95% CI = [1.42, 4.53]), who perceived the intervention as relevant for trachoma prevention (AOR = 7.78, 95% CI = [4.38, 13.82]), and who had received training in F and E components of the SAFE strategy (AOR = 2.17, 95% CI = [1.24, 3.78]), were

**Table 1. Cross tabulation of characteristics of the health workers; Western Province, April 2023 (N = 388).**

| Factor | Frequency (%) | Adopted F and E hygiene promotion n (%) | Did not adopt F and E hygiene promotion n (%) | P-values (Chi-square/Fisher's Exact/ Wilcoxon Rank sum) |
|---|---|---|---|---|
| **Sex** | | | | |
| Male | 191 (49.23) | 91 (47.64) | 100 (52.36) | 0.9896c |
| Female | 197 (50.77) | 94 (47.72) | 103 (52.28) | |
| **Age (Median, IQR)** | 40 (34, 50) | 41 (34, 50) | 40 (34,49) | 0.687wr |
| **District** | | | | |
| Kalabo | 163 (42.01) | 96 (58.90) | 67 (41.10) | **0.001c** |
| Kaoma | 165 (42.53) | 65 (39.39) | 100 (60.61) | |
| Shangombo | 60 (15.46) | 24 (40.00) | 36 (60.00) | |
| **Type of Facility** | | | | |
| Health Post | 135 (34.79) | 70 (51.85) | 65 (48.15) | 0.141c |
| Rural Health Centre | 200 (51.55) | 96 (48.00) | 104 (52.00) | |
| Urban Health Centre | 53 (13.66) | 19 (35.85) | 34 (64.15) | |
| **Supervisor Position** | | | | |
| Clinician | 120 (30.93) | 49 (40.83) | 71 (59.17) | **<0.0001c** |
| Environmental Health T/O | 172 (44.33) | 101 (58.72) | 71 (41.28) | |
| Other | 96 (24.74) | 35 (36.46) | 61 (63.54) | |
| **Transport availability** | | | | |
| Not available | 279 (71.91) | 108 (38.71) | 171 (61.29) | **<0.0001c** |
| Rarely available | 41 (10.57) | 26 (63.41) | 15 (36.59) | |
| Readily available | 68 (17.53) | 51 (75.00) | 17 (25.00) | |
| **Perceived complexity of F and E hygiene promotion** | | | | |
| Not complicated | 252 (64.95) | 136 (53.97) | 116 (46.03) | **0.001c** |
| Complicated | 136 (35.05) | 49 (36.03) | 87 (63.97) | |
| **Perceived relevance of F and E hygiene promotion** | | | | |
| Relevant | 202 (52.06) | 138 (68.32) | 64 (31.68) | **<0.0001c** |
| Not relevant | 186 (47.94) | 47 (25.41) | 138 (74.58) | |
| **Training on F and E Promotion** | | | | |
| Trained | 170 (43.81) | 94 (55.29) | 76 (44.71) | **0.008c** |
| Not trained | 218 (56.19) | 91 (41.74) | 127 (58.26) | |
| **IEC material availability** | | | | |
| Available | 126 (32.47) | 74 (58.73) | 52 (41.27) | **0.003c** |
| Not available | 262 (67.53) | 111 (42.37) | 151 (57.63) | |
| **Attitude- (There is need to conduct F and E activities if MDA has been conducted)** | | | | |
| Very positive | 129 (33.25) | 69 (53.49) | 60 (46.51) | **<0.0001c** |
| Positive | 93 (23.97) | 49 (52.69) | 44 (47.31) | |
| Neutral | 41 (10.57) | 25 (60.98) | 16 (39.02) | |
| Negative | 83 (21.39) | 20 (24.10) | 63 (75.90) | |
| Very negative | 42 (10.82) | 22 (52.38) | 20 (47.62) | |

c- Pearson chi$^2$ test, f = Fisher's exact test, wr = Wilcoxon rank-sum test, IEC = Information, education and communication material, MDA = Mass drug administration.

significantly more likely to adopt the intervention. Further, health workers were significantly more likely to adopt the intervention if they had a very positive attitude (AOR = 3.12, 95% CI = [1.39, 6.99]), a neutral attitude (AOR = 7.29, 95% CI = [2.71, 19.58]), or a positive attitude (AOR = 3.73, 95% CI = [1.57, 8.90]) towards conducting facial and environmental

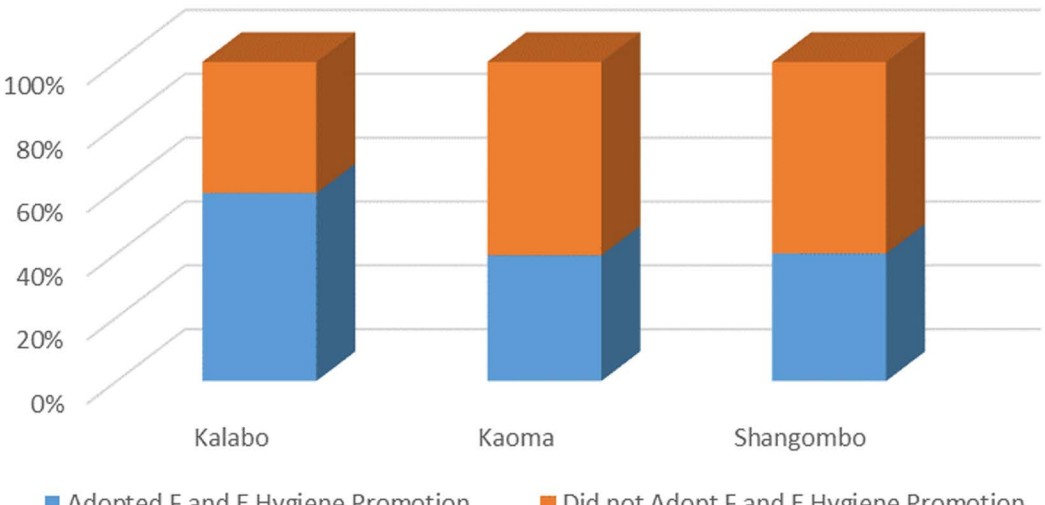

**Fig 1. Adoption of F and E hygiene promotion by district. The blue portions of the bars represent the proportions of health workers who adopted the intervention whereas the orange segments represent those who did not.**

hygiene promotion in communities where mass drug administration (MDA) had recently occurred. Availability of information, education, and communication (IEC) materials was also associated with higher adoption (AOR = 3.04, 95% CI = [1.69, 5.46]). On the other hand, this study found that there was no significant association between adoption of the intervention and sex, type of facility and supervisor's position.

## Discussion

This study sought to determine the level of, and factors associated with adoption of facial and environmental hygiene promotion among health workers. The study found that slightly less than half of the participants (47.68%) adopted the intervention and of the three districts included in the study, Kalabo had the highest proportion of participants who had adopted the intervention and Shangombo had the lowest proportion. The factors associated with adoption of facial and environmental hygiene promotion included district of respondent, transport availability, perceived complexity of the intervention, perceived relevance of the intervention, training, availability of IEC materials and attitude towards facial and environmental hygiene promotion activities in communities that have recently undergone MDA.

The low level of adoption of facial and environmental hygiene promotion among health workers may be inadequate to significantly advance population level goals of reducing trachoma prevalence, potentially contributing to the slow progress towards elimination in Western Province. This concern is supported by previous studies showing that alongside administration of antibiotics, regions with enhanced adoption and implementation of facial and environmental hygiene promotion achieve faster control of trachoma [22,23]. In contrast to our finding, Ghana, one of the first countries to eliminate trachoma, implemented all components of the SAFE strategy, including F and E promotion, with equal and high level intensity [24–26]. Our findings are consistent with previous literature that also reported low adoption of the F and E components [8,16], underscoring a persistent gap that may be hindering effective trachoma control.

Both the core components of the intervention were only moderately adopted, except in Shangombo District, where adoption of community distribution of IEC material was higher. These findings suggest that facial and environmental

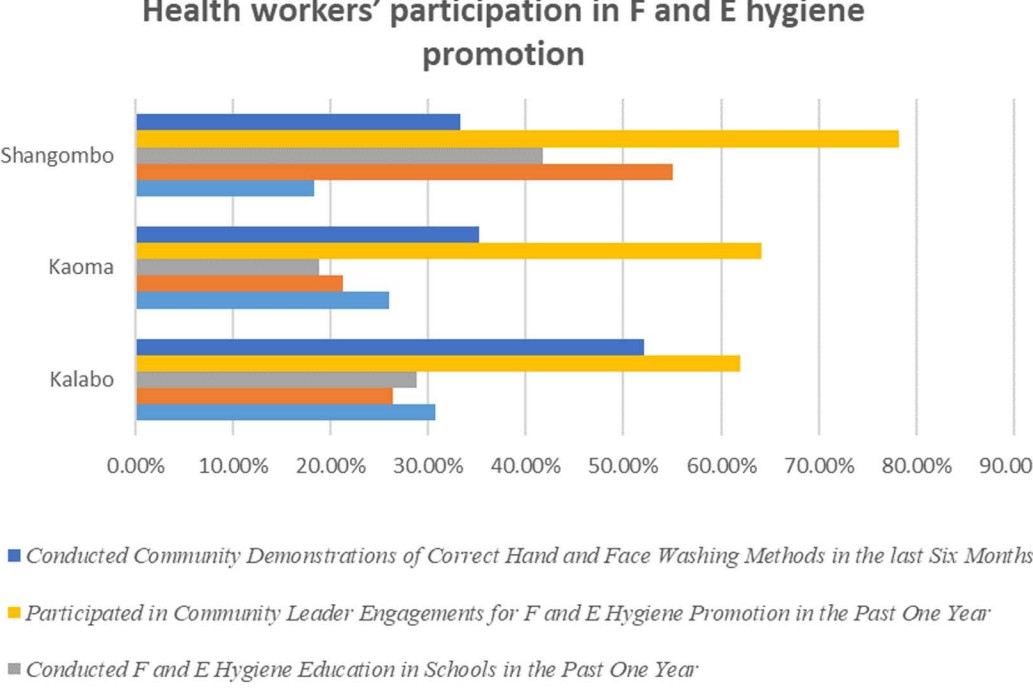

**Fig 2. Health workers' participation in F and E hygiene promotion.** The yellow bars represent health workers who engaged community leaders for facial and environmental hygiene promotion; dark blue bars indicate those who conducted correct hand and face washing methods demonstrations; gray bars show those who provided facial and environmental hygiene education in schools; orange bars represent participation in community IEC material distribution; and light blue bars indicate attendance at stakeholder consultation meetings.

hygiene promotion may not be implemented in accordance with the WHO recommendations. WHO and other trachoma stakeholders encourage countries implementing the SAFE strategy to scale up behavioral change strategies for health promotion such as community demonstrations of correct hand and face washing techniques [7,27]. Similarly, the review of grey literature on facial cleanliness and environmental improvement interventions for trachoma control found that very few interventions targeted behavior change activities [27]. This may be due to a lack of research on proximal precedents of behavior such as attitudes and norms which hinders the design of behavioral change strategies. The systematic review [27], which included studies from diverse settings, strengthens the likelihood that our findings reflect broader trends across other SAFE-implementing contexts.

Health workers from Kaoma and Shangombo districts were less likely to adopt facial and environmental hygiene promotion compared to those from Kalabo. This observation could have been influenced by attitudes towards facial and environmental hygiene promotion. Most of the participants with positive attitude towards the need to conduct facial and environmental hygiene promotion in communities which have recently undergone mass drug administration of antibiotics were from Kalabo compared to the other two districts. This is supported by previous studies which reported that attitude was an important factor in the adoption and implementation of health education and promotion activities [28,29].

Transport availability for use in health promotion activities affected adoption of facial and environmental hygiene promotion. Majority of the health workers reported not having transport for use in community health promotion activities,

**Table 2. Factors associated with adoption of facial and environmental hygiene promotion (Unadjusted and adjusted logistic regression results).**

| Explanatory Variable | Unadjusted Estimates | | | Adjusted Estimates | | |
|---|---|---|---|---|---|---|
| | OR | (95%CI) | P-value | AOR | (95%CI) | P-value |
| **Sex** | | | | | | |
| Female | – | – | – | | | |
| Male | 1.00 | (0.67, 0.49) | 0.989 | | | |
| **Age** | 1.002 | (1.000, 1.003) | **0.042** | | | |
| **District** | | | | | | |
| Kalabo | – | – | – | – | – | – |
| Kaoma | 0.45 | (0.29, 0.71) | **<0.0001** | 0.39 | (0.20, 0.76) | **0.006** |
| Shangombo | 0.47 | (0.25, 0.85 | **0.013** | 0.36 | (0.16, 0.81) | **0.014** |
| **Type of health facility** | | | | | | |
| Health Post | – | – | **–** | – | – | **–** |
| Rural Health Centre | 0.86 | (0.55, 1.33) | 0.490 | 0.57 | (0.32, 1.04) | 0.066 |
| Urban Health Centre | 0.52 | (0.27, 1.01) | 0.050 | 0.62 | (0.23, 1.72) | 0.362 |
| **Supervisor's position** | | | | | | |
| Clinician | – | – | – | – | – | – |
| Environmental Health Officer | 2.06 | (1.28, 3.31) | **0.003** | 1.65 | (0.85, 3.20) | 0.136 |
| Other | 0.83 | (0.48, 1.45) | 0.513 | 0.47 | (0.21, 1.02) | 0.055 |
| **Facility transport type** | | | | | | |
| Bicycle | – | – | – | – | – | – |
| Motorcycle | 0.59 | (0.28, 1.24) | 0.165 | 0.45 | (0.16, 1.30) | 0.140 |
| Motor vehicle | 1.69 | (0.50, 5.70) | 0.397 | 4.18 | (1.14, 15.3) | **0.031** |
| None | 0.37 | (0.22, 0.61) | **<0.0001** | 0.65 | (0.32, 1.32) | 0.232 |
| **Transport availability** | | | | | | |
| Not available | – | – | – | – | – | – |
| Rarely available | 2.74 | (1.39, 5.42) | **0.004** | 1. 96 | (0.73, 5.27) | 0.182 |
| Readily available | 4.75 | (2.61, 8.66) | **<0.0001** | 3.06 | (1.38, 6.80) | **0.006** |
| **Perceived intervention complexity** | | | | | | |
| Complicated | – | – | – | – | – | – |
| Not complicated | 2.08 | (1.36, 3.20) | **0.001** | 2.54 | (1.42, 4.53) | **0.002** |
| **Perceived intervention relevance** | | | | | | |
| Not relevant | – | – | **–** | – | – | **–** |
| Relevant | 6.38 | (4.09, 9. 95) | **<0.0001** | 7.78 | (4.38, 13.82) | **<0.0001** |
| **Training on F and E** | | | | | | |
| Not trained | – | – | – | – | – | – |
| Trained | 1.73 | (1.15, 2.59) | **0.008** | 2.17 | (1.24, 3.78) | **0.006** |
| **IEC material** | | | | | | |
| Not available | – | – | – | – | – | – |
| Available | 1.94 | (1.26, 2.98) | **0.003** | 3.04 | (1.69, 5.46) | **<0.0001** |
| **Attitude- (There is need to conduct F and E activities if MDA has recently been conducted)** | | | | | | |
| Negative | – | – | – | – | – | – |
| Positive | 3.51 | (1.84. 6.71) | **<0.0001** | 3.73 | (1.57, 8. 90) | **0.003** |
| Neutral | 4.92 | (2.20, 11.01) | **<0.0001** | 7.29 | (2.71, 19.58) | **<0.0001** |
| Very Negative | 3.47 | (0.88, 7.61) | 0.402 | 4.60 | (1.50, 14.12) | 0.008 |
| Very positive | 3.62 | (1.79, 6.67) | **<0.0001** | 3.12 | (1.39, 6. 99) | **0.006** |

Adjusted estimates are showing results of an investigator led backward stepwise multivariable logistic regression, OR = Unadjusted odds ratio, AOR =Adjusted odds ratio.

and they were less likely to adopt the intervention compared to those who indicated that transport was rarely or readily available. This finding is not surprising as lack of funding and resources such as transport make it difficult for health workers to carry out health promotion activities [30]. Further, our findings align with those of a previous study which found that organizations implementing facial cleanliness and environmental improvement interventions often received less funding and support, for resources such as transport, compared to those focusing on surgery and antibiotics components [15].

The perceived complexity and relevance of facial and environmental hygiene promotion affected adoption of this intervention. Health workers who found the intervention hard to implement were less likely to adopt it compared to those who felt it was uncomplicated and easy to execute. Similarly, health workers who perceived the intervention as being relevant for trachoma prevention were more likely to adopt it compared to those who perceived it otherwise. This may be because individuals who perceive an activity as complicated or unimportant are less likely to be motivated to engage in it. These finding are consistent with a previous study in Australia, which found that school respondents had limited knowledge on trachoma, its prevalence and the role of facial cleanliness and environmental improvements in trachoma prevention, resulting in reduced confidence in teaching about hygiene and trachoma prevention practices [28]. Similarly, a study found that believing that IEC material were relevant in the delivery of health education and promotion activities was associated with higher odds of material utilization [31].

IEC materials are a critical component of health promotion activities as they make it easy to deliver health education information to the community and may motivate staff to participate in health promotion activities. Our study found that participants who had IEC materials for facial and environmental hygiene promotion available were more likely to adopt the intervention compared to those who did not. This finding could be because IEC materials inform health workers and provide a guidance on key messages to include in sensitization talks, making it easy for them to conduct health promotion activities. This, in turn, increases their motivation to engage in such activities. This finding is consistent with findings from previous research which showed that availability of IEC materials affected workers' delivery of health education and promotion activities [32,33].

Training of health workers is important to enhance the delivery of facial and environmental hygiene promotion activities [28]. Our study found that health workers who had received training in facial cleanliness and environmental improvements components were more likely to adopt the intervention compared to those who had not. This could be because those who have been trained are more likely to be knowledgeable and confident to carry out the intervention activities. This finding is consistent with another study that found that lack of training was one of the factors which was perceived to influence the implementation of health education and promotion activities by health workers [29].

We found that Shangombo District had not widely adopted community demonstrations of correct hand and face washing methods. Similarly, a study assessing trachoma elimination progress in districts with persistent trachoma in Western Province, Zambia, reported that the prevalence of TF in children aged 1–9 years only declined from 17.9% to 12.5% between 2018 and 2023 in Shangombo [13]. This indicates that trachoma remains a public health problem in the district, meeting the WHO criteria for persistent trachoma [13]. These findings suggest that failure to implement behavior change interventions, such as community demonstrations of proper hand and face washing methods, may contribute to continued transmission of active trachoma.

The study findings can support program implementers in lobbying for dedicated funding for facial and environmental hygiene promotion from government and donor agencies, given their commitment to implementing the full SAFE strategy in the study area. Such funds could be used to ensure the availability of key resources including IEC materials, transport and to facilitate training for health workers on the importance of facial and environmental hygiene promotion in trachoma elimination.

This study had two limitations. Firstly, using a stratified random sampling design potentially introduced clustering of the data at the different levels of the cluster and strata variables, signifying that our findings may not be generalizable outside

the three districts of Zambia. To mitigate this, we report robust estimates of odds ratios and confidence intervals to counter clustering effects. Further, we incorporated the strata and cluster variables into the model to avoid underestimating the confidence intervals. Secondly, the use of a questionnaire to gather self-reported information on adoption of facial and environmental hygiene promotion is subject to recall bias. Participants could possibly recall wrongly that they had taken part in intervention activities when in fact they had not.

## Conclusion

Overall, health worker adoption of facial and environmental hygiene promotion was low. Key factors influencing adoption included the availability of transport and IEC materials, attitudes towards facial and environmental hygiene promotion, and perceptions of the intervention's relevance and complexity. Health workers with easy access to transport and IEC materials were more likely to adopt the intervention, highlighting the need for consistent resource availability. This can be addressed through increased funding and budget allocation for facial and environmental hygiene promotion activities, especially given the Government of Zambia's commitment to implementing all components of the SAFE strategy. Additionally, health workers who had received training in F and E promotion were more likely to adopt the intervention than those who had not, underscoring the importance of targeted capacity building. Training should emphasize the relevance of facial and environmental hygiene promotion in trachoma prevention, provide clear guidance on how to implement the intervention, and address perceptions of complexity. Importantly, training should also reinforce the need to continue facial and environmental hygiene promotion even in communities where mass drug administration of antibiotics for trachoma has already been conducted. These insights should inform strategies to improve adoption of facial and environmental hygiene promotion, ultimately supporting population-level behavior change and advancing progress towards trachoma elimination.

## Supporting information

**S1 Data. The data set used for analysis.** This Excel file contains anonymized raw data collected from the questionnaires, which was used for all statistical analyses in the manuscript.
(XLS)

## Acknowledgments

Martha Kasongo is a recipient of a TDR scholarship under the Postgraduate Training Scheme in Implementation Research at the University of Zambia. We are grateful for the support for the training scheme provided by UNICEF, the UNDP, the World Bank and the WHO Special Program for Research and Training in Tropical Diseases (TDR)

## Author contributions

**Conceptualization:** Martha Kasongo, Choolwe Jacobs, Silumbwe Adam, Patricia Maritim, Hikabasa Halwindi.

**Data curation:** Martha Kasongo.

**Formal analysis:** Martha Kasongo.

**Investigation:** Martha Kasongo.

**Methodology:** Martha Kasongo, Hikabasa Halwindi.

**Supervision:** Choolwe Jacobs, Hikabasa Halwindi.

**Writing – original draft:** Martha Kasongo.

**Writing – review & editing:** Martha Kasongo, Choolwe Jacobs, Silumbwe Adam, Patricia Maritim, Joseph Mumba Zulu, Hikabasa Halwindi.

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
