## [Decision Letter · Decision Letter 0]

9 Jul 2025

Response to Reviewers
Revised Manuscript with Track Changes
Manuscript

Shaden Kamhawi

co-Editor-in-Chief

Paul Brindley

co-Editor-in-Chief

**Additional Editor Comments:**
**Journal Requirements:**

2) We have noticed that you have uploaded Supporting Information files, but you have not included a list of legends. Please add a full list of legends for your Supporting Information files after the references list.

**Reviewers' comments:**

**Key Review Criteria Required for Acceptance?**

**Methods:**

-Are the objectives of the study clearly articulated with a clear testable hypothesis stated?

-Is the study design appropriate to address the stated objectives?

-Is the population clearly described and appropriate for the hypothesis being tested?

-Is the sample size sufficient to ensure adequate power to address the hypothesis being tested?

-Were correct statistical analysis used to support conclusions?

-Are there concerns about ethical or regulatory requirements being met?

Reviewer #1: In Background (page 2), the authors didn't explain about the reason of the "interventions particularly for facial cleanliness and environmental improvement have been sub-optimally adopted"

In Introduction (page 6), the authors have to explain, in depth, the usefulness of the results. The authors only mentioned: "The results will inform......through the identified determinants of adoption"

In Methods (page 6), the study employed the RE-AIM/PRISM but it's not clear if there some evidence of the use of RE-AIM/PRISM in trachoma's studies

The authors also mentioned the following: "it is one of the understudied areas in this field". I recommend to explain the idea in introduction

In sampling and participant recruitment (page 8) stratified random sampling method was used so it's necessary to estimate the sample size calculation with design effect

I suggest to explain a little more some variables like training, position held, length of service, etc before the results. e.g.1. perceived intervention complexity, how it complexity define?

e.g.2. transport availability, and the options are not available, rarely available and readily available. what does mean rarely? once, twice a month?

In data collection (page 9), the authors mentioned that the quantitative data was collected by the principal investigator with help of 3 research assistants. I suggest to introduce more information about the research assistants, e.g. they trained before the study's execution

Reviewer #2: The main issue is that the sampling strategy was confusing. Is this a random sample?

Reviewer #3: The objectives of the study are clearly articulated with a clear testable hypothesis stated

The study design is appropriate to address the study objectives

The targeted and sampled population is clearly described and appropriate for the hypothesis being tested

The sample size is sufficient to ensure adequate power to address the hypothesis being tested

The study applied the correct statistical analysis to support conclusions

The study has no concerns about ethical or regulatory requirements. All ethical or regulatory requirements were met

**Results:**

-Does the analysis presented match the analysis plan?

-Are the results clearly and completely presented?

-Are the figures (Tables, Images) of sufficient quality for clarity?

Reviewer #1: I don't understand why the authors show the characteristics of the participants in two tables. Is there some reason?

I recommend to write the main results (page 10). e.g. 34.79% were from health posts. I will change this instead 51.55 were from rural health centre

Reviewer #2: Yes

Reviewer #3: The analysis which was conducted matches the analysis plan

The study results are clearly and fully presented

The figures (Tables, Images) included in the study are sufficient quality for clarity

**Conclusions:**

-Are the conclusions supported by the data presented?

-Are the limitations of analysis clearly described?

-Do the authors discuss how these data can be helpful to advance our understanding of the topic under study?

-Is public health relevance addressed?

Reviewer #1: The discussion is too long and it's not clear about the information that the authors want to give to the readers.

e.g. Our findings are similar to those of Tsang et al (2021) perhaps due to the fact that both research were conducted in rural areas and the study populations were very similar (page 22).

The majority of the studies about trachoma were developed in rural setting so the information gave for the authors is not relevant

Reviewer #2: Generally yes though in the limitations paragraph I would comment that these results may not be generalizable outside these districts of Zambia, and I would limit any conclusions only to the study area and not try to extrapolate to other settings.

Reviewer #3: The conclusions are supported by the data presented, though the conclusion should be strengthened and rephrased to focus on key information from the results presented

The limitations of analysis should be clearly described

The authors have discussed how these data can be helpful to advance our understanding of the topic under study

Public health relevance has been addressed

**Editorial and Data Presentation Modifications?**

Reviewer #1: I suggest only use the variables that are really important to the study. I think there are so many variables that it could create a confusion to the reader

Reviewer #2: (No Response)

Reviewer #3: The authors should make minor revisions

**Summary and General Comments:**

Reviewer #1: I will review the regression analysis carefully, as the authors used stratified random sampling. This sampling design requires adjustments in the regression model to avoid underestimating the confidence intervals. Maybe, Incorporate strata and cluster variables into the model.

Reviewer #2: Line 33: what intervention? It’s not clear what these percentages refer to

Line 34: not sure the lower odds in Kaoma and Shangombo need to be in the abstract. That seems like knowledge that would be important for local policymakers but this paper is presumably being written to share more broadly generalizable knowledge. I would delete that sentence. (And probably the final sentence of the conclusion)

Line 40: similar comment, but need to specify what adoption is, and how it is defined. Is this self-reported adoption? What components were adopted?

Line 80: “core components”: is this promoted by the government in some official way? Is this a more informal assessment by the authors based on activities of NGOs? Please more clearly state how these core components have been defined.

Line 81: Are IEC materials a standardized set of materials made by the government? Or by an NGO? Or does this just mean any type of materials that an individual health center might have made.

Line 136: “full SAFE strategy”: who makes this decision / who implements this. Is this from the government/MOH? Is this NGOs?

Line 156: the methods for stratified random sampling were not clear. In line 160, does “representative” = “random” (and if so, please use the word random). I wondered if 24 health facilities were randomly selected from the total pool of 52 health facilities using probability proportional to size sampling, and then if within these 24 selected facilities systematic sampling was employed. (but no details are given about how the systematic sampling was actually done, and my summary above is not actually stated anywhere but just my hypothesis of what was done.) Please be more explicit about the sampling.

Line 222: how was “knowledgeable about facial and environmental hygiene promotion” defined? Is this knowledge? Or familiarity?

Line 225: how was a “positive attitude” defined, or how was the question asked? (If as stated in Table 1b, I would include this in the text as well)

Table 1b: “perceived relevance of F and E”: relevance for what? Relevant for trachoma control? Relevant as an activity that the health center should be doing?

Table 1b: what does “incentives” mean. That the workers are paid extra to do hygiene promotion above and beyond their salary?

Table 1c: spell out full words of IEC in footnote or caption

Table 2: I believe the AOR was constructed from all variables in the table after applying a backwards stepwise approach. (A) what was the p-value threshold to remove a variable from the model? (maybe include this in the methods) (B) in the table caption or footnote I would include that the adjusted model was selected using a backwards stepwise approach. (C) in the table caption please specify what the outcome is for the ORs (adoption of promotion activities I believe?).

Line 288: “determinants” makes this sound like a causal relationship. But here only an association has been found. I would change to “The main factors associated with…” or something like that. Also: I did not scour the text after noticing this, but please read through and make sure that the text uses words that expresses only a cross-sectional association, and not a causal relationship.

Line 391: trachoma prevalence was not reported here. A citation should be given, and also a little more detail. (What does a “decrease in trachoma prevalence” mean? Between which years? According to which surveys? Does this mean TF?)

Line 299: “there is a need to increase funding”: I am not sure I agree. If I were a donor I would want evidence that implementing F&E is effective. Currently there are very few interventional studies that have shown F&E to be effective. Almost all the evidence comes from observational studies. If health workers are not adopting hygiene promotion but hygiene promotion does not work, then why should funders devote more money to it? Also, the results of this one study may not be relevant to other places. You could consider some caveats to this sentence, eg: “The results of this study demonstrate a need for increased funding and training in facial and environmental hygiene promotion in the study area, given the district/zone/federal stated plans/commitment/goals for hygiene programming.” Or something like that, which would be accurate for the Zambian setting.

Fig 1: these are petty comments, and if it is very difficult to change I would not worry about it, but for your next paper consider: (a) no need for any decimal places on y axis; (b) third dimension on bar graphs conveying no information and thus should be dropped.

Reviewer #3: Under key words – consider rephrasing to Determinants; Adoption; Facial and environmental hygiene promotion; SAFE strategy; Trachoma elimination; Western Province, Zambia as key words

Line 22 – consider including this statement; In Zambia, Western province is the most affected province with trachoma. Ref this in the main body1-3Reference this work to justify selection of Western province for this research

1. Mwale C, Mboni C, Saasa N, et al.; Assessing trachoma elimination progress in districts with persistent trachoma, Western Province, Zambia. International health 2025:ihaf041.

2. Mwale C, Mumbi W, Funjika M, et al.; Prevalence of trachoma in 47 administrative districts of Zambia: results of 32 population-based prevalence surveys. Ophthalmic epidemiology 2018;25(sup1):171-180.

3. Mwale C, Mulamba M, Chama E, et al.; Assessment of outcomes of the patients undergoing surgery for Trachomatous Trichiasis, Western Province, Zambia. Medical Journal of Zambia 2025;52(3):332-339.

Line 1 – edit to read Determinants of adoption of facial and environmental hygiene promotion in the ‘SAFE strategy’ for trachoma elimination in Western Province, Zambia – a cross-sectional study or

Determinants of adoption of facial and environmental hygiene promotion in the ‘SAFE strategy’ for trachoma elimination in Western Province, Zambia

Line 22 – Rephrase to The SAFE strategy (Surgery, Antibiotics, Facial Cleanliness 23 and Environmental Improvement) is recommended by WHO for elimination of trachoma.

Line 52 – elimination of trachoma in endemic communities

Line 63 – elimination program officers and stakeholders

Ine 72, include references 1 and 2 above

Line 102 - help trachoma elimination program implementers identify possible solutions to enhance adoption

Line 152 - equation, n ≥ (Z2 p [1-p])/e2 where, n is the sample size, z is the selected critical value of desired – state the value of each parameter used in the equation

Line 147 - Sampling – where facility in charges included among participants, if not why?

Line 389 - This study found that Shangombo, a district which did not widely adopt community demonstration for correct face and hand washing methods did not record a significant reduction in trachoma prevalence. – ref 1 above

Line 392 interventions may lead to persistent active trachoma – ref 1 above

Line 398 – what is MDA, which appears in the conclusion

PLOS authors have the option to publish the peer review history of their article (what does this mean? ). If published, this will include your full peer review and any attached files.

**Do you want your identity to be public for this peer review?** For information about this choice, including consent withdrawal, please see our Privacy Policy .

Reviewer #1: No

Reviewer #2: No

Reviewer #3: No

**Figure resubmission:****Reproducibility:** To enhance the reproducibility of your results, we recommend that authors of applicable studies deposit laboratory protocols in protocols.io, where a protocol can be assigned its own identifier (DOI) such that it can be cited independently in the future. Additionally, PLOS ONE offers an option to publish peer-reviewed clinical study protocols. Read more information on sharing protocols at https://plos.org/protocols?utm_medium=editorial-email&utm_source=authorletters&utm_campaign=protocols

---

## [Editor Report · Decision Letter 1]

18 Aug 2025

Dear Ms. Kasongo,

We are pleased to inform you that your manuscript 'Factors associated with health worker adoption of facial and environmental hygiene promotion in the ‘SAFE strategy’ for trachoma elimination  in Western Province, Zambia.' has been provisionally accepted for publication in PLOS Neglected Tropical Diseases.

Best regards,

Josh M Colston, Ph.D.

Academic Editor

Amy Morrison

Section Editor

Shaden Kamhawi

co-Editor-in-Chief

Paul Brindley

co-Editor-in-Chief

Thank you for your resubmission, and congratulations on an excellent study and article that is well worthy of publication in PLOS NTDs.

---

## [Editor Report · Acceptance letter]

Dear Kasongo,

We are delighted to inform you that your manuscript, " 

Factors associated with health worker adoption of facial and environmental hygiene promotion in the ‘SAFE strategy’ for trachoma elimination in Western Province, Zambia.," has been formally accepted for publication in PLOS Neglected Tropical Diseases.

Best regards,

Shaden Kamhawi

co-Editor-in-Chief

Paul Brindley

co-Editor-in-Chief
